# Nationwide Prevalence and Outcomes of Long-Term Nasogastric Tube Placement in Adults

**DOI:** 10.3390/nu14091748

**Published:** 2022-04-22

**Authors:** Chung Y. Hsu, Jung-Nien Lai, Woon-Man Kung, Chao-Hsien Hung, Hei-Tung Yip, Yu-Chen Chang, Cheng-Yu Wei

**Affiliations:** 1Graduate Institute of Clinical Medical Science, China Medical University, Taichung 40402, Taiwan; hsucy63141@gmail.com; 2School of Chinese Medicine, College of Chinese Medicine, China Medical University, Taichung 40402, Taiwan; ericlai111@gmail.com; 3Department of Chinese Medicine, China Medical University Hospital, Taichung 40447, Taiwan; 4Department of Exercise and Health Promotion, College of Kinesiology and Health, Chinese Culture University, Taipei 11114, Taiwan; nskungwm@yahoo.com.tw; 5Department of Neurology, Feng Yuan Hospital, Taichung 42055, Taiwan; mithradil@gmail.com; 6Management Office for Health Data, Clinical Trial Center, China Medical University Hospital, Taichung 40447, Taiwan; fionyip0i0@gmail.com; 7College of Medicine, China Medical University, Taichung 40402, Taiwan; 8Show Chwan International Dementia and Movement Disorder Center, Chang Bing Show Chwan Memorial Hospital, Changhua County 50544, Taiwan; turquoise654@gmail.com; 9Department of Neurology, Chang Bing Show Chwan Memorial Hospital, Changhua County 50544, Taiwan

**Keywords:** long-term, nasogastric tube, tube feeding, dysphagia, prevalence, outcome

## Abstract

Tube feeding (TF) is commonly used for patients with severe swallowing disturbance, and patients with chronic dysphagia are often provided with a long-term nasogastric tube (NGT). However, nationwide epidemiological data on long-term NGT placement are limited. The present study identified the prevalence and outcomes of patients with long-term NGT placement in Taiwan. Data were obtained from the Longitudinal Health Insurance Database. Patients with NGT placement for more than 3 months between 2000 and 2012 were enrolled in this cohort study. An NGT cohort of 2754 patients was compared with 11,016 controls matched for age, sex, residential area, and comorbidities. The prevalence rate of long-term NGT reached 0.063% in 2005 and then remained stable at 0.05–0.06%. The major causes of NGT placement were stroke (44%), cancer (16%), head injury (14%), and dementia (12%). Men (63%) were more likely to have long-term NGT placement than women (37%). The adjusted hazard ratios were 28.1 (95% CI = 26.0, 30.3) for acute and chronic respiratory infections; 26.8 (95% CI = 24.1, 29.8) for pneumonia, 8.84 (95% CI = 7.87, 9.93) for diseases of the esophagus, stomach, and duodenum; and 7.5 (95% CI = 14.7, 20.8) for mortality. Patients with NGT placement for more than 6 months had a higher odds ratio (1.58, 95% CI = 1.13, 2.20) of pneumonia than those with NGT placement for less than 6 months. Only 13% and 0.62% of the patients underwent rehabilitation therapy and percutaneous endoscopic gastrostomy, respectively. Long-term NGT use was associated with a higher risk of comorbidities and mortality. Stroke was the main illness contributing to long-term NGT use. Further interventions are necessary to improve the negative effects of long-term TF.

## 1. Introduction

Long-term tube feeding (TF) is essential for the nutritional support of patients with chronic swallowing difficulties. In Western and a few Asian countries (e.g., Japan and Korea), enteral tube feeding (ETF) is widely used in patients requiring long-term TF [1,2,3]. However, in many Asian countries, patients with chronic and severe dysphagia are commonly provided with a long-term nasogastric tube (NGT) [4,5,6]. Although NGTs resolve the physical problem of intake, they are associated with higher morbidities and mortality as well as the increased costs of long-term care [7,8,9].

In Taiwan, 29.2% of residents in long-term care facilities are fed using an NGT [10]. In the United States, 34% of patients with dementia in nursing homes require TF [11]. In Germany, the average rate of percutaneous endoscopic gastrostomy (PEG) in nursing homes is 6.6% [12], and in Japan, the mean proportions of ETF in long-term care facilities, rehabilitation hospitals, nursing homes, and sanatorium medical facilities are 7.4%, 7.9%, 11.6%, and 36.3%, respectively [2]. In Israel, 26% of older people with advanced dementia require feeding tubes, 13% using PEG and 13% with NGT placement [13]. 

Epidemiological research on TF has mostly focused on long-term care settings and special groups or regions in one country. Therefore, this national population study aimed to establish a comprehensive epidemic insight into patients with long-term NGT placement.

## 2. Materials and Methods

### 2.1. Data Source

In 1995, Taiwan launched a compulsory insurance program and established a database to enable evidence-driven interventions. In this study, we used the Longitudinal Health Insurance Database (LHID), constituting one million randomly selected patients, for analysis. The data included outpatient records, inpatient records, medications, and treatment history. The disease codes used were those of the International Classification of Diseases, Ninth Revision, Clinical Modification (ICD-9 CM). For privacy, all identifying data were encrypted. This study was approved by the Institutional Review Board of the China Medical University Hospital Research Ethics Committee (CMUH104-REC2-115 (AR-4)).

### 2.2. Study Population

Patients with NGT placement for more than 3 months between 2000 and 2012 were the target population in this cohort study. The health insurance procedure code for NGT is 47017C. Patients without NGT placement were selected as controls. The index date for the case patients was the first date of NGT placement and that of the controls was a random date between 2000 and 2012. Four controls were matched to a case patient according to sex, age, residential area, and comorbidities through propensity score matching. Patients were excluded if they were younger than 20 years old and had had an NGT placement for less than 3 months. Figure 1 provides a flowchart of participant selection. The main causes of NGT placement included stroke (ICD-9 CM codes 430–438), head injury (ICD-9 CM codes 850–854 and 959.01), Parkinson’s disease (PD) or Parkinsonism (ICD-9 CM codes 332 and 333, excluding 333.1–333.8), dementia (ICD-9 CM codes 290.0, 290.1, 290.2, 290.3, 294.1, and 331.0). The covariates used were hypertension (ICD-9 CM codes 401–405), diabetes mellitus (DM, ICD-9 CM code 250), hyperlipidemia (ICD-9 CM code 272), coronary artery disease (CAD, ICD-9 CM codes 410–414), congestive heart failure (CHF, ICD-9 CM code 428), chronic kidney disease (CKD, ICD-9 CM code 585), chronic obstructive pulmonary disease (COPD, ICD-9 CM codes 490–492, 494, and 496), and atrial fibrillation (AF, ICD-9 CM code 427.31).

### 2.3. Outcome Measurements

The outcomes developed over 1 year after NGT placement included acute and chronic respiratory infections (ICD-9 CM codes 460–519); acute respiratory infections (ICD-9 CM codes 460–466); pneumonia (ICD-9 CM codes 480–486); diseases of the esophagus, stomach, and duodenum (ICD-9 CM codes 530–537); and mortality. For patients with NGT placement, we also considered their rate of rehabilitation and subsequent acceptance of PEG.

### 2.4. Statistical Analysis

We calculated the prevalence of patients with NGT placement for more than 3 months for each year by dividing the numbers of events by the total population. We then investigated each main cause of NGT placement. The difference in demographic variables and baseline comorbidities between the case and control cohorts was examined using a chi-square test. The hazard ratios (HRs) and 95% confidence intervals (CIs) were estimated using a Cox proportional hazard model and adjusted for sex, age, residential area, and comorbidities. A logistic regression model was applied to evaluate the odds ratio (OR) to identify associations between the length of NGT placement and pneumonia. The cumulative incidence curve was plotted using the Kaplan–Meier method and a log-rank test. All the statistical analyses were performed using SAS software (version 9.4; SAS Institute, Cary, NC, USA), and the significance level was *p* < 0.05. 

## 3. Results

The number of patients with long-term NGT placement of more than 3 months increased annually from 680,208 in 2000 to 759,734 in 2012 (Figure 2); the prevalence rate of NGT placement increased from 0.025% in 2000 to 0.063% in 2005. After 2005, the prevalence rate remained stable at 0.05–0.06%. Finally, we enrolled 2754 patients in the NGT cohort and 11,016 subjects in the non-NGT control cohort. Table 1 presents the major causes of NGT placement: stroke (44%), cancer (16%), head injury (14%), dementia (12%), and PD and Parkinsonism (5%). Men (63%) were more likely than women (37%) to have NGT placement. The mean age of patients was 73.4 ± 14.2 years, and 78% were older than 65 years. With regard to geographical distribution, 38%, 30%, 19%, and 13% of the patients were in Northern, Central, Eastern (including outlying islands), and Southern Taiwan, respectively. Most of them had high levels of comorbidities, which included hypertension (78%), COPD (50%), DM (40%), hyperlipidemia (36%), CHF (16%), CAD (12%), CKD (12%), and AF (9%).

Table 2 summarizes the baseline characteristics of the patients in the two cohorts with the distribution after matching. The rates of patients with long-term NGT placement undergoing rehabilitation therapy or PEG were only 13% and 0.62%, respectively. Despite rehabilitation therapy, the failure rate in terms of NGT removal was as much as 80%.

The data in Table 3 indicate that all outcomes increased in patients with long-term NGT placement compared with the controls. The adjusted HRs for acute and chronic respiratory infections, acute respiratory infections, pneumonia, diseases of the esophagus, stomach, and duodenum, and mortality were 28.1 (95% CI = 26.0, 30.3), 6.16 (95% CI = 4.73. 8.04), 26.8 (95% CI = 24.1, 29.8), 8.84 (95% CI = 7.87, 9.93), and 17.5 (95% CI = 14.7, 20.8), respectively. Figure 3 demonstrates that the 12-month cumulative incidence curves for each outcome were significantly higher in the NGT cohort than in the non-NGT cohort (log-rank test, *p* < 0.001). The patients with NGT placement for more than 6 months had a higher OR (1.58, 95% CI = 1.13, 2.20) for pneumonia than those with NGT placement for less than 6 months (Table 4).

## 4. Discussion

Swallowing disorders are common; according to a recent big-data survey, they affect one in six adults [14]. Results on dysphagia prevalence differ not only depending on age and etiology but also due to methodology. Overall, epidemiological reports have indicated that dysphagia is more common (6–50%) among older adults [15,16]. In our population-based study focused on long-term NGT placement, the prevalence rate was approximately 0.05–0.06% in the adult population, and the rate was dominant in men and increased in line with age, which is consistent with other studies [14,15,16]. This is the first longitudinal follow-up study to reveal the annual prevalence in an adult population with long-term NGT placement. As the global population continues to age, more attention should be paid to the management of TF in long-term care.

The main causes of long-term NGT placement differ depending on age. In our study, the most common cause in young adults (20–40 years) was head injury, and stroke was the most common cause in those older than 41 years; cancer was the second most common cause. Incidence of dysphagia following head injury was reported to be 27–30% [17]. The predictors of dysphagia resulting from stroke lesions were medullary (OR = 6.2, 95% CI = 1.5–25.8), insular (OR = 4.8, 95% CI = 2.0–11.8), and pontine (OR = 3.6, 95% CI = 1.2–10.1) predictors, followed by brain atrophy (OR = 3.0, 95% CI = 1.04–8.6), internal capsular lesions (OR = 2.9, 95% CI = 1.2–6.6), and increasing age (OR = 1.4, 95% CI = 1.1–1.8) [18]. Notably, degenerative diseases, including dementia (12%) and PD and Parkinsonism (5%), were the key etiology of long-term NGT placement in older adults. Dysphagia was reported in 11–81% of patients with PD, 7–29% of patients with Alzheimer disease, and 19–57% of patients with frontotemporal dementia [15,17]. Stroke, cancer, and head injury prevention could substantially reduce long-term TF.

One study observed that most nursing home residents requiring TF accepted long-term NGT placement (97.2%) rather than PEG (2.8%) [19]. In the present study, only 0.62% of patients received PEG after long-term NGT placement (Table 2). Although two systemic reviews have demonstrated the superiority of ETF over NGT placement [20,21], most patients still accept NGT placement for long-term TF in Taiwan. One of these studies concluded that PEG decreased failures in TF and gastrointestinal bleeding; PEG had a greater effect on food delivery and albumin concentration [20]. The other study also concluded that PEG reduced failures in TF and was safer and more effective than NGT placement [21]. Limited PEG use could be explained using the four “A” domains: acceptability, availability, affordability, and accountability. A lower acceptability of PEG is influenced primarily by traditional Chinese culture and the concept of end-of-life body integrity. PEG availability is also limited because of the lack of special PEG teams, and affordability is lacking due to insufficient subsidies from the National Health Insurance Administration; furthermore, the higher accountability of PEG and the need for more complex follow-up care also reduce the performance of PEG [19]. In brief, limited PEG use could be attributed primarily to affordability.

However, scholars have different opinions on the outcomes of ETF and NGT placement. A Cochrane review revealed no significant difference in mortality and morbidities between ETF and NGT groups [21]. Another systematic review did not support the use of PEG over NGT placement in patients with no stroke [22]. One randomized controlled trial (RCT) including patients with stroke (63.5%) and no stroke (36.5%) revealed greater 4-month survival in the PEG group, but no differences were detected in nutrition outcomes, 1-year hospitalization, pneumonia, or mortality between the NGT and PEG groups [6]. Another RCT verified that early PEG feeding increased disability and mortality in patients with acute stroke [23]. A further retrospective observational study demonstrated that patients with direct ETF after acute stroke had greater disability, complications, and mortality compared with temporary NGT placement alone [24]. For patients with terminal head and neck cancer, a prospective observational study revealed longer hospitalization in the NGT group but no difference in the quality of life and functional status between NGT and PEG groups [25]. Focusing on patients with dementia or psychiatric diseases, a retrospective study revealed that patients with TF survived longer than those without TF and PEG was safer than NGT placement [26]. The data-supported indications for PEG include cancer, stroke, amyotrophic lateral sclerosis, multiple sclerosis, and severe brain damage; dementia is classified as a doubtful indication [27,28]. Physicians must consider individual circumstances and optimize professional decisions when patients require long-term TF.

In this study, only 13% of patients with long-term NGT placement received rehabilitation therapy, and the success rate of NGT removal was only 20%. The main cause was the extremely insufficient speech–language pathologists (SLPs). SLPs, as we all know, are the most qualified providers for dysphagia services. In Taiwan, the certified SLPs officially began from 434 (1.87 per 100,000 population) in 2010 and increased to 1256 (5.40 per 100,000 population) in 2020 [29]. Compared with the United States, the certified SLPs were up to 147,470 (44.2 per 100,000 population) in 2021 [30]. A multidisciplinary approach is therefore required for long-term TF diagnosis and management. The clinical team should include neurologists, gastroenterologists, geriatricians, rehabilitation physicians, ear–nose–throat specialists, radiologists, dietitians, nutritionist, SLPs, nurses, and caregivers. Team members should receive relevant training, including in terms of screening tests, clinical evaluation, and instrumental assessments. In addition, swallowing training, nutritional interventions, and newer stimulation techniques would be beneficial for patients [31,32]. However, such integrated teams have not yet formed a task organization for long-term NGT removal in most hospitals in Taiwan. Supportive government policies and public advocacy are necessary for developments in long-term TF care.

A strength of this study is its use of a big, national-level data set. However, the study also has several limitations. First, because it was limited to LHID data, the research enrollment period was 2001–2012, and the end of follow-up date was 31 December 2012. Further studies should use data covering a longer period and with longer follow-up periods. Second, causal inference was precluded by this study’s retrospective cohort research design, and randomized trials should be conducted in the future. Third, because the data were anonymized, data on key lifestyle or baseline variables, such as smoking habits, body mass index, lifestyle, and family history, could not be obtained. Fourth, relevant clinical data, such as those pertaining to symptoms and signs or obtained from imaging results, pathology reports, and laboratory data, were unavailable in the database. However, records related to NGT placement, PEG, and diagnoses were reliable. Further prospective observational and transnational studies are necessary to support the findings.

This nationwide longitudinal follow-up and retrospective study revealed the prevalence and outcomes of long-term NGT placement in adults. Higher comorbidities and mortality in relation to lower usage of PEG and rehabilitation therapy were revealed in the NGT cohort. Our findings suggest that governments, long-term care facilities, and health authorities should pay greater attention to patients with dysphagia.

## Figures and Tables

**Figure 1 nutrients-14-01748-f001:**
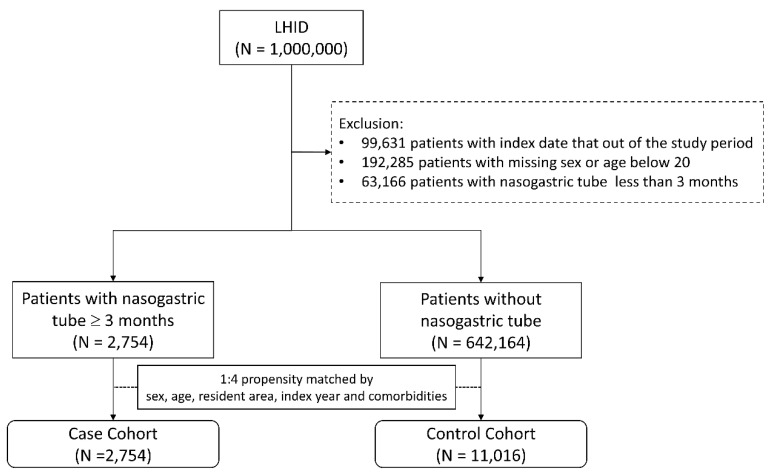
Flowchart for study population selection. LHID: Longitudinal Health Insurance Database.

**Figure 2 nutrients-14-01748-f002:**
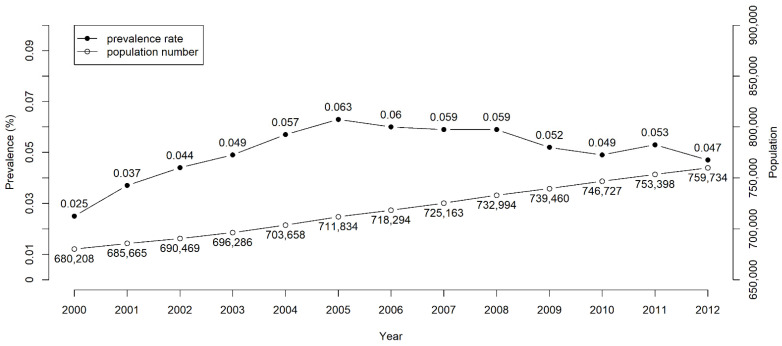
Prevalence rates and numbers of long-term nasogastric tube placement in Taiwan from 2000 to 2012.

**Figure 3 nutrients-14-01748-f003:**
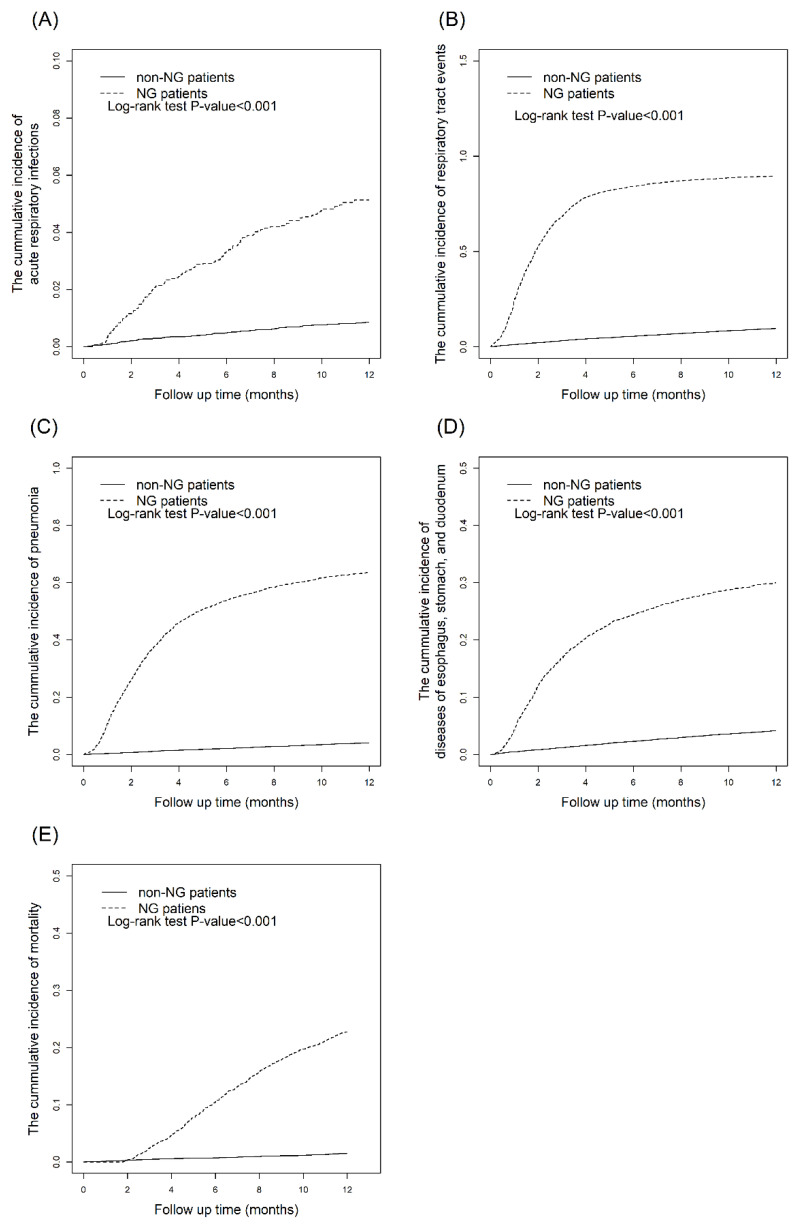
Kaplan–Meier curves for the cumulative incidences of outcomes within one year in NGT placement cohort and non-NGT placement cohort: (**A**) Acute and chronic respiratory infections. (**B**) Acute respiratory infections. (**C**) Pneumonia. (**D**) Diseases of esophagus, stomach, and duodenum. (**E**) Mortality.

**Table 1 nutrients-14-01748-t001:** The characteristics of patients with nasogastric tube placement with different main cause.

Characteristics	NGT Patients	Main Cause					
		Stroke	Head Injury	PD and Parkinsonism	Dementia	Cancer	Something Else
Number	2754	1202 (44%)	396 (14%)	141 (5%)	324 (12%)	444 (16%)	310 (11%)
Gender							
Female	1009 (37%)	463 (39%)	126 (32%)	49 (35%)	131 (40%)	139 (31%)	127 (41%)
Male	1745 (63%)	739 (62%)	270 (68%)	92 (65%)	193 (60%)	305 (69%)	183 (59%)
Age							
20–30	45 (2%)	6 (0.5%)	28 (7.1%)	0 (0%)	0 (0%)	6 (1.4%)	7 (2.3%)
31–40	46 (2%)	11 (0.9%)	16 (4.0%)	1 (0.7%)	0 (0%)	12 (2.7%)	8 (2.6%)
41–50	127 (5%)	48 (4.0%)	20 (5.1%)	2 (1.4%)	1 (0.3%)	44 (9.9%)	13 (4.2%)
51–65	380 (14%)	198 (17%)	58 (15%)	7 (5.0%)	3 (0.9%)	77 (17%)	41 (13%)
>65	2156 (78%)	939 (78%)	274 (69%)	131 (93%)	320 (99%)	305 (69%)	241 (78%)
Mean, (SD)	73.4 (14.2)	73.5 (12.5)	68.6 (18.4)	77.8 (9.35)	81.8 (7.41)	70.3 (15.5)	73.2(15.2)
Area							
Northern	1042 (38%)	478 (40%)	124 (31%)	46 (33%)	133 (41%)	174 (39%)	104 (34%)
Central	832 (30%)	384 (32%)	146 (37%)	33 (23%)	76 (24%)	123 (28%)	91 (29%)
Southern	367 (13%)	146 (12%)	43 (11%)	33 (23%)	50 (15%)	61 (14%)	48 (15%)
Eastern and island	513 (19%)	194 (16%)	83 (21%)	29 (21%)	65 (20%)	86 (19%)	67 (22%)
Comorbidities							
Hypertension	2135 (78%)	1023 (85%)	266 (67%)	122 (87%)	278 (86%)	299 (67%)	198 (64%)
Diabetes	1113 (40%)	532 (44%)	141 (36%)	62 (44%)	131 (40%)	165 (37%)	107 (35%)
Hyperlipidemia	983 (36%)	474 (39%)	136 (34%)	67 (47%)	123 (38%)	136 (31%)	73 (24%)
CAD	325 (12%)	151 (13%)	40 (10%)	20 (14%)	46 (14%)	43 (9.7%)	32 (10%)
CHF	448 (16%)	178 (15%)	39 (9.8%)	33 (23%)	78 (24%)	69 (16%)	64 (21%)
CKD	333 (12%)	142 (12%)	33 (8.3%)	22 (17%)	43 (13%)	58 (13%)	39 (13%)
COPD	1387 (50%)	549 (46%)	175 (44%)	91 (65%)	218 (67%)	225 (51%)	155 (50%)
AF	244 (9.0%)	121 (10%)	15 (3.8%)	16 (11%)	43 (13%)	26 (5.9%)	25 (8.1%)
Hospital level							
Medical center	952 (35%)	455 (38%)	136 (34%)	37 (26%)	74 (23%)	184 (41%)	93 (30%)
District hospital	1209 (44%)	513 (43%)	202 (51%)	60 (43%)	133 (41%)	161 (36%)	142 (46%)
Local hospital	593 (22%)	234 (19%)	58 (15%)	44 (31%)	117 (36%)	99 (22%)	75 (24%)

NGT: nasogastric tube; PD: Parkinson’s disease; SD: standard deviation; CAD: coronary artery disease; CKD: chronic kidney disease; CHF: congestive heart failure; COPD: chronic obstruction pulmonary disease; AF: atrial fibrillation.

**Table 2 nutrients-14-01748-t002:** The baseline characteristics of patients with and without nasogastric tube placement.

	Non-NGT		NGT			Rehabilitation		PEG	
	N = 11,016		N = 2754			N = 364 (13%)		N = 7 (0.62%)	
	*n*	%	*n*	%	*p*-Value	*n*	%	*n*	%
Gender					0.74				
Female	3996	36%	1009	37%		118	(32%)	5	(29%)
Male	7020	64%	1745	63%		246	(68%)	12	(71%)
Age, year					1.00				
20–30	180	2%	45	2%		17	(5%)	0	(0%)
31–40	185	2%	46	2%		14	(4%)	1	(6%)
41–50	494	4%	127	5%		33	(9%)	1	(6%)
51–65	1520	14%	380	14%		97	(27%)	4	(24%)
>65	8637	78%	2156	78%		203	(56%)	11	(65%)
Area					0.61				
Northern	4239	38%	1042	38%		155	(43%)	9	(53%)
Central	3314	30%	832	30%		162	(45%)	5	(29%)
Southern	1454	13%	367	13%		27	(7%)	3	(18%)
Eastern and island	2009	18%	513	19%		20	(5%)	0	(0%)
Comorbidities									
Hypertension	8637	78%	2135	78%	0.33	265	(73%)	11	(65%)
Diabetes	4552	41%	1113	40%	0.98	138	(38%)	7	(41%)
Hyperlipidemia	3855	35%	983	36%	0.51	152	(42%)	4	(24%)
CAD	1100	10%	325	12%	0.01	36	(10%)	0	(0%)
CKD	1732	16%	448	16%	0.50	38	(10%)	0	(0%)
CHF	1180	11%	333	12%	0.04	17	(5%)	0	(0%)
COPD	5704	52%	1387	50%	0.19	123	(34%)	5	(29%)
AF	782	7%	244	9%	0.002	30	(8%)	3	(18%)
Main cause									
Stroke			1202	44%		237	(65%)	6	(35%)
Head injury			396	14%		86	(24%)	1	(6%)
PD and parkinsonism			141	5%		9	(2%)	1	(6%)
Dementia			324	12%		6	(2%)	3	(18%)
Cancer			444	16%		20	(5%)	6	(35%)
Something else			310	11%		12	(3%)	0	(0%)
Hospital level									
Medical center			952	35%		199	(55%)	9	(53%)
District hospital			1209	44%		149	(41%)	5	(29%)
Local hospital			593	22%		16	(4%)	3	(18%)
Re-intubation of NGT						291	(80%)		

NGT: nasogastric tube; PEG: percutaneous endoscopic gastrostomy; CAD: coronary artery disease; CKD: chronic kidney disease; CHF: congestive heart failure; COPD: chronic obstruction pulmonary disease; AF: atrial fibrillation; TBI: traumatic brain injury; PD: Parkinson’s disease.

**Table 3 nutrients-14-01748-t003:** The outcomes within one year.

	Non-NGT			NGT			HR	
	*n*	PY	IR	*n*	PY	IR	cHR (95% CI)	aHR ^†^ (95% CI)
Morbidities								
Acute and chronic respiratory infections	1061	10,375	1.02	2444	734	33.3	25.4 (23.5,27.4) ***	28.1 (26.0,30.3) ***
Acute respiratory infections	94	10,860	0.09	131	2389	0.55	6.20 (4.72,8.03) ***	6.16 (4.73,8.04) ***
Pneumonia	444	10,706	0.41	1701	1392	12.2	25.6 (23.0,28.5) ***	26.8 (24.1,29.8) ***
Diseases of esophagus, stomach, and duodenum	456	10,680	0.43	788	1962	4.02	8.70 (7.78,9.81) ***	8.84 (7.87,9.93) ***
Mortality	162	10,912	0.15	625	2464	2.54	17.3 (14.6,20.6) ***	17.5 (14.7,20.8) ***

NGT: nasogastric tube; PY: person-year; IR: incidence rate per 10 person-year; HR: hazard ratio; cHR: crude hazard ratio; aHR: adjusted hazard ratio; ***: *p*-value < 0.001; †: adjusted for sex, age, resident area, and comorbidities.

**Table 4 nutrients-14-01748-t004:** The association of time span of nasogastric tube placement and pneumonia.

Time Span of NGT	N	Pneumonia	COR (95% CI)	AOR ^†^ (95% CI)
3 months–6 months	2573	1572	1.00 (reference)	1.00 (reference)
>6 months	181	129	1.57 (1.13,2.20) **	1.58 (1.13,2.20) **

NGT: nasogastric tube; cOR: crude odds ratio; aOR: adjusted odds ratio; **: *p*-value < 0.01; †: adjusted for sex, age, resident area, and comorbidities.

## Data Availability

Data sharing is not applicable to this article.

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
