# Peer review of "Nationwide Prevalence and Outcomes of Long-Term Nasogastric Tube Placement in Adults"

_nutrients, 2022, doi:10.3390/nu14091748_

Round 1
Reviewer 1 Report
Really good topic to discuss. As we are seeing a lot of people are going through NG tube placement in hospitals and it seems this topic has addressed those issues.Author Response
Thank you for your precious comments.

Reviewer 2 Report
In this manuscript (nutrients-1676903), the authors identified the prevalence and outcomes of patients with long-term NGT placement in Taiwan and showed the higher odds ratio of pneumonia in them. Furthermore, they suggested the relationship between lower usage of PEG and rehabilitation therapy and higher comorbidities and mortality. This MS seems to explain the actual circumstances in Taiwan, but I have some concerns about this MS:
Major comments
The authors said that only 13% of patients with long-term NGT placement received rehabilitation therapy and the success rate of NGT removal was only 20%. I am interested in the reason for the low rate of rehabilitation therapy in patients with long-term NGT placement. Are the medical system or medical facilities in Taiwan poorly equipped to provide swallowing rehabilitation therapy or have a shortage of skilled team members?
Please introduce the criteria for the application of PEG in Taiwan if there is. In some western countries, PEG cannot be applied to patients with dementia.
Please discuss the possibility that introducing PEG will reduce the comorbidities and mortality citing the previous studies.
Author Response
Thank you for your precious comments.
- We have mentioned the cause and revised by reviewer’s comments accordingly as‟ In this study, only 13% of patients with long-term NGT placement received rehabilitation therapy, and the success rate of NGT removal was only 20%. The main cause was the extremely insufficient speech-language pathologists (SLPs). SLPs, as we all know, are the most qualified providers for dysphagia services. In Taiwan, the certified SLPs officially began from 434 (1.87 per 100,000 population) in 2010 increased to 1256 (5.40 per 100,000 population) in 2020 [29]. Compared with the United States, the certified SLPs were up to 147,470 (44.2 per 100,000 population) in 2021 [30]. A multidisciplinary approach is therefore required for long-term TF diagnosis and management. The clinical team should include neurologists, gastroenterologists, geriatricians, rehabilitation physicians, ear–nose–throat specialists, radiologists, dietitians, nutritionist, SLPs, nurses, and caregivers. Team members should receive relevant training, including in terms of screening tests, clinical evaluation, and instrumental assessments. In addition, swallowing training, nutritional interventions, and newer stimulation techniques would be beneficial for patients [31,32]. However, such integrated teams have not yet formed a task organization for long-term NGT removal in most hospitals in Taiwan. Supportive government policies and public advocacy are necessary for developments in long-term TF care. ” in the fifth paragraph of the discussion
- In Taiwan, the criteria of PEG are still based on international standards or guideline. We have revised by reviewer’s comments accordingly as‟ The data-supported indications for PEG include cancer, stroke, amyotrophic lateral sclerosis, multiple sclerosis and severe brain damage; dementia is classified as a doubtful indication [27,28]. Physicians must consider individual circumstances and optimize professional decisions when patients require long-term TF.” in the fourth paragraph of the discussion section.
- Several comparisons of the outcomes including comorbidities and mortality between PEG and NTG have been discussed in the fourth paragraph of the discussion section.

Reviewer 3 Report
In this study, big data was used to summarize the results of a fact-finding survey on long-term tube feeding, especially NGT, which was very interesting. I have nothing to point out regarding the research method, the structure and content of the dissertation. However, there are some suspected typographical errors, so please correct them.
Line49, 228,230,266
Is the description of "NTG" an error of "NGT"?
If it is an abbreviation for another term, it needs explanation.
Line115-128
“The Materials and Methods should be described ... provided approval and the corresponding ethical approval code.”
Is these sentences insertion of the results of peer review? Delete it if you don't need it.
Author Response
Thank you for your precious comments.
- We have corrected the error of “NTG” to “NGT” in the whole article.
- We have provided approval and the corresponding ethical approval code. We have revised by reviewer’s comments accordingly as‟ In 1995, Taiwan launched a compulsory insurance program and established a database to enable evidence-driven interventions. In this study, we used the Longitudinal Health Insurance Database (LHID), constituting one million randomly selected patients, for analysis. The data included outpatient records, inpatient records, medications, and treatment history. The disease codes used were those of the International Classification of Diseases, Ninth Revision, Clinical Modification (ICD-9-CM). For privacy, all identifying data were encrypted. This study was approved by the Institutional Review Board of the China Medical University Hospital Research Ethics Committee (CMUH104-REC2-115 (AR-4)).” in the Materials and Methods paragraph of the discussion section.
- We have deleted some unnecessary sentences by reviewer’s comments.

Round 2
Reviewer 2 Report
In this manuscript (nutrients-1676903), the authors revised the first version. These revisions may be accessible to many readers. I judge that this MS generally reaches the publication standard.